# SAW Humidity Sensing with rr-P3HT Polymer Films

**DOI:** 10.3390/s24113651

**Published:** 2024-06-05

**Authors:** Wiesław Jakubik, Jarosław Wrotniak, Cinzia Caliendo, Massimiliano Benetti, Domenico Cannata, Andrea Notargiacomo, Agnieszka Stolarczyk, Anna Kaźmierczak-Bałata

**Affiliations:** 1Institute of Physics CSE, Silesian University of Technology, 44-100 Gliwice, Poland; anna.kazmierczak-balata@polsl.pl; 2Institute of Electronics, Silesian University of Technology, 44-100 Gliwice, Poland; jaroslaw.wrotniak@polsl.pl; 3Institute for Photonics and Nanotechnologies, IFN-CNR, 00133 Rome, Italy; cinzia.caliendo@cnr.it (C.C.); andrea.notargiacomo@cnr.it (A.N.); 4Institute of Microelectronics and Microsystems, CNR, 00133 Rome, Italy; massimiliano.benetti@cnr.it (M.B.); domenico.cannata@cnr.it (D.C.); 5Department of Physical Chemistry and Technology of Polymers, Silesian University of Technology, 44-100 Gliwice, Poland; agnieszka.stolarczyk@polsl.pl

**Keywords:** surface acoustic wave, SAW, relative humidity, poly-3-hexylthiophene, rr-P3HT, polymer films

## Abstract

In the present paper the humidity sensing properties of regioregular rr-P3HT (poly-3-hexylthiophene) polymer films is investigated by means of surface acoustic wave (SAW) based sensors implemented on LiNbO_3_ (128^0^ Y-X) and ST-quartz piezoelectric substrates. The polymeric layers were deposited along the SAW propagation path by spray coating method and the layers thickness was measured by atomic force microscopy (AFM) technique. The response of the SAW devices to relative humidity (rh) changes in the range ~5–60% has been investigated by measuring the SAW phase and frequency changes induced by the (rh) absorption in the rr-P3HT layer. The SAW sensor implemented onto LiNbO_3_ showed improved performance as the thickness of the membrane increases (from 40 to 240 nm): for 240 nm thick polymeric membrane a phase shift of about −1.2 deg and −8.2 deg was measured for the fundamental (~78 MHz operating frequency) and 3rd (~234 MHz) harmonic wave at (rh) = 60%. A thick rr-P3HT film (~600 nm) was deposited onto the quartz-based SAW sensor: the sensor showed a linear frequency shift of ~−20.5 Hz per unit (rh) changes in the ~5–~50% rh range, and a quite fast response (~5 s) even at low humidity level (~5% rh). The LiNbO_3_ and quartz-based sensors response was assessed by using a dual delay line system to reduce unwanted common mode signals. The simple and cheap spray coating technology for the rr-P3HT polymer films deposition, complemented with fast low level humidity detection of the tested SAW sensors (much faster than the commercially available Michell SF-52 device), highlight their potential in a low-medium range humidity sensing application.

## 1. Introduction

Humidity measurement is critical in many branches of human activities such as industrial processes, agricultural productions, room and motor vehicles conditions, production of medicines as well the very precise research experiments. The most of the contemporary humidity sensors are based on the capacitance, resistance, impedance, acoustic and optical methods, however, their mutual drawback is often the low accuracy and hysteresis effect at the low humidity detection levels [1,2,3,4,5,6,7]. Considerable efforts have thus been focused on the development of highly sensitive materials and novel device structures to address these issues [8,9,10,11,12,13,14,15,16]. Because of very high surface to volume ratio and specific physical/chemical properties, application of nanomaterials such as graphene oxides, carbon nanotubes, electrospun nanofibers, metal oxide nanowires, and various polymer structures in humidity sensors has resulted in many exciting progresses over the recent years [17,18,19,20,21,22,23,24,25,26,27,28]. The development of novel sensitive materials for applications in high-performance humidity sensors, especially for specific industry applications (which very often require fast, high accuracy, low cost and low level humidity sensing) deserve more investigation and analyses.

High-performance sensors can be achieved by utilizing the Surface Acoustic Waves (SAW)-based devices: since the acoustic wave energy is confined to a thin near-surface region of the propagation medium, the SAW devices are highly sensitive to any surface perturbations that take place at the specific sensitive layer placed on the top of the piezoelectric substrate and have a fast response. Examples of external stimuli which can perturb the SAW velocity and attenuation can be surface mass density, electrical conductivity, or elastic parameters changes, to cite just a few. The SAW sensors sensitivity can be improved by using high operating frequency devices: thus, devices based on harmonic waves can lead to enhanced operation frequency of the SAW devices without requiring expensive nanofabrication techniques necessary to reduce the metal electrode dimensions. The transducers used in the present article efficiently excite both the fundamental and 3rd harmonic Rayleigh wave, while the fifth and seventh harmonics, although excited as well, suffer large propagation loss and hence are not suitable for further sensing tests. Consequently, the LiNbO_3_-based sensors were tested at both the fundamental and third harmonic wave.

The sensor design is a key component of all the sensing devices and greatly affect the response and recovery time, the signal-to-noise ratio, the detection limit, and so on. In the case of SAW devices, the sensing layer is the main core of the SAW sensor, as it acts as the interface between the environment to be tested and the SAW: the target molecules are adsorbed on the surface of the sensing membrane thus inducing the SAW velocity change due to the mass loading effect. For these purposes, the selection of a highly sensitive and selective sensing membrane is desired for the SAW sensors.

In this paper the new thin film sensor structures of rr-P3HT (regio-regular poly-3-hexylthiophene) polymer, prepared by means of simple and cheap spray coating method, are investigated in SAW single and dual-delay line configuration systems for the low-medium relative humidity range. The results proved the ability of a low (~5%) RH detection with quite fast responses (~5 s), much faster than commercially available Michell SF-52 dew point devices.

## 2. Materials and Methods

The tested SAW sensors consisted in two different basic devices: a single delay line implemented onto 128^0^ Y-X LiNbO_3_ and a dual delay line implemented on ST-x quartz. The operating frequency of the former was ~78 MHz for the fundamental SAW frequency and ~234 MHz for the third harmonic, while that of the latter was ~205 MHz. Figure 1a,b show the two devices with the rr-P3HT sensitive thin film deposited in between the two interdigital transducers (IDTs) by means of a simple and low cost spray coating method. One of the two delay lines onto quartz was intentionally left bare to act as a reference element. The quartz-based dual delay lines were commercial devices (from SAW Components, Dresden, Germany) with wavelength of ~15 µm, number of fingers/side N = 365 and IDTs centre to centre distance L = 5853.5 µm. Whereas conventional photolithography and lift-off techniques were employed to pattern the interdigital transducers (IDTs) onto an Al layer 1500 Å thick grown onto the LiNbO_3_ substrate by rf sputtering technique from a high purity Al target in Ar atmosphere. The IDTs have split finger configuration: each IDT consists of 80 electrodes with a periodicity of 80 μm; the acoustic aperture was equal to 1568 μm and the IDTs center-to-center distance was 6600 μm. The LiNbO_3_-based SAW devices were assembled on a printed circuit board (PCB) and the solder pads were electrically bounded to SMA connectors to measure the scattering parameter (S_21_) with a vector network analyzer (Keysight P9371A, Santa Rosa, CA, USA). The quartz-based SAW devices were connected to a switchable generator to collect the frequency signals from the sensing (delay line covered by the film) and reference (delay line with the bare surface) devices. Figure 1c shows the schematic diagram of the experimental set up which includes the VNA, the mass flow controller (MFC), the bubble humidification system, and the dry and wet air mixing system.

The rr-P3HT polymer membrane was deposited in between the two IDTs of the single line and dual-delay line by means of the simple spray coating method. The investigated polymer films were deposited from a previously prepared solution, by spraying with a pistol of a nozzle thickness of ~0.4 mm. Compressed synthetic air at a pressure of about 1 atm was applied as a carrier gas. The solution was prepared by dissolving ~1 mg of the rr-P3HT polymer “snowflakes” in 1 mL of chloroform. The nozzle distance from the substrate with IDTs during the spray process was approx. 40 mm. Experimentally suited to the utilized masking window and the parameters of spray coating, a distance of less than ~40 mm causes (for this scale for a SAW substrate) too strong a blow of the atomized solution on the transducer, which can cause the agent to penetrate under the mask and settle on the IDTs, resulting in its permanent damage. A distance of more than 40 mm leads to unnecessary and lossy covering of a larger area of the mask with a solution of rr-P3HT polymer. The deposition time was approx. 2–4 s. The procedure allowed to form the polymer films with different thicknesses, estimated on the base of AFM measurements [29]. The most important steps of this simple method are presented in the Figure 2.

## 3. Results

### 3.1. AFM Characterization of the Films

The morphology of the rr-P3HT films was investigated by means of Atomic Force Microscopy (AFM) with a tapping mode. The results for the polymer films onto LiNbO_3_ with an estimated thicknesses of ~240, ~130, ~80 and 40 nm are depicted in Figure 3, while those of the thicker film (~600 nm) onto quartz is presented in Figure 4.

Figure 3 and Figure 4 show AFM images data of the surface topography of the rr-P3HT layers deposited on LiNbO_3_ and quartz substrates. Figure 3a–d shows the 2D AFM maps collected at a scan-size of 5 μm referred to samples having thicknesses of ~240, ~130, ~80 and 40 nm, respectively, which were used in films in a single delay line configuration. The corresponding Rs surface roughness values are 30 nm, 28 nm, 10 nm, and 5 nm. The AFM topography of a much thicker film (~600 nm) used in a dual-delay line quartz module for the more precise frequency investigations, is reported in Figure 4 showing the 3D rendering of the morphology. The Rs value for this sample is ~94 nm. The AFM data revealed that the surface roughness of the layers increases with increasing the layer thickness. The morphology data of Figure 3 and Figure 4 show that the roughness is mainly due to the characteristic features of the thin spray-coated films. In fact, circular holes or pits are present, which are likely originated by the evaporation of the solvent present in the film as it is being deposited. The lateral size of such structures is in the range between few hundreds of nanometers and about 1 μm, while the depth is of the order of 10 nm. These structures are clearly less present and pronounced for the thinnest film (~40 nm).

### 3.2. Sensing Results

In the single delay line configuration on LiNbO_3_ substrate, the measured phase shifts of the 3rd harmonic SAW was almost seven times greater than that of the fundamental wave (−8.2 deg versus −1.2 deg) for the sensing film ~240 nm thick at 60% RH, what is presented in Figure 5. Therefore, we decided to limit the sensor characterization by focusing on the third harmonic wave for further investigation. The higher order harmonics, although excited by the IDTs, were not used to characterize the sensor due to their small amplitudes, as shown in Appendix A.

Figure 6 and Figure 7 show the LiNbO_3_-based SAW sensors response to the humidity for the rr-P3HT polymer film ~240 nm thick at 20% and 40% rh. Each measure is repeated three times: the average phase shifts for the three independent measuring tests of the SAW 3rd harmonic were estimated on the values ~−1.9 deg for 20% rh and ~−3.9 deg for the 40% rh.

The example of the humidity sensing results for the rr-P3HT polymer film ~240 nm thick at 60% RH in single SAW delay line configurations is showed in Figure 8. Here, the average phase shifts for the three independent measuring tests of the SAW 3rd harmonic was ~−7.9 deg.

The bare reference acoustic delay line (without the polymer film) was also exposed to the dry-wet-dry air cycles and it showed a response much smaller than that of the structure with the polymer film. In the bare case, only the mass loading effect takes place due to the lack of the active polymer film as well as to the electrical dipole interactions of the polar water molecules with the electrical field associated to the SAW on the free substrate surface (which will be the subject of the future research). The exemplary results of such an interaction are presented in Figure 9.

Therefore, the (rh) induced phase shifts of the third harmonic SAW in rr-P3HT polymer films on the LiNbO_3_ substrate are presented in Table 1; the relative phase shifts (absolute phase shift—phase shift of the reference free line) are shown in Table 2 and in Figure 10.

Based on the experimental results listed in Table 1 and Table 2, it can be noted that the phase signals decrease as the thickness of the polymer layer decreases. The interactions with humidity of the studied rr-P3HT polymer films are found to be lower in the case of thinner films: this result is compatible with what has been published in the literature on humidity sensitivity measurements of SAW sensors based on thin and thick polymer films of PVP and PVA [30]. In this article, SAW sensors coated with thicker polymer films showed broader responses with good resolution, demonstrating that film thickness is a key factor in achieving good resolution and broad response in humidity sensing. Consequently, for the development of the next phase of the research (study of the double-delay line configuration on quartz), a polymer film was spin-deposited thick enough (~600 nm) (Figure 4) to allow the excitation and detection of the SAW, and such as to offer good sensor response.

Figure 11 shows, as an example, the time response (the frequency shifts vs time curves) of the dual delay line configuration on quartz for five different relative humidity levels. The first response at rh = 5.3% was about −200 Hz (which was not detected by the professional humidity sensor Michell SF52 with a dew point (dp) temperature detector); the subsequent sensor responses were ~−290 Hz (rh~8.3%), ~−420 Hz (rh~16%), ~−510 Hz (rh~19%) and ~−1180 Hz (rh~52%). The base line for a dp = −19.9 °C is for rh ~5.3%. The long-term frequency drift is caused by the slightly increasing temperature of the quartz module (green curve).

Figure 12 shows the calibration curve (frequency shift vs RH) of the quartz-based dual delay line with rr-P3HT film ~600 nm thick: the sensor sensitivity, estimated by the slope of the curve, resulted to be of −20.5 Hz/% rh. The limit of detection (LOD), i.e., the smallest amount of relative humidity capable of producing a sensor response distinguishable from noise, can be theoretically estimated at ~0.4% which is an admissible value considering that a frequency shift of 20 Hz is easily measurable even in the presence of noise with an amplitude of 8 Hz, as measured in our case.

## 4. Discussion

The rr-P3HT polymer films deposited by means of spray coating method were not perfectly uniform in the thickness, as confirmed by AFM measurements. The coating method used is simple and inexpensive, however, its biggest disadvantage is the lack of thickness uniformity; however, the deposition method allows obtaining films with a repeatable average thickness. By comparing the responses of the sensors it was observed that thicker films produce higher responses than thinner films: this effect can be attributed to the large number of absorption centers of water molecules present in thicker films. On the other hand, the larger phase shifts observed for the 3rd harmonic SAW in LiNbO_3_, in comparison to the 1st one, was attributed to the larger operating frequency which enables higher sensitivity due to the mass loading effect, as opposed to the acoustoelectric effect which is not effective versus the frequency in the single layer/substrate sensor configuration [31,32,33,34,35].

In the quartz dual-delay line configurations covered by the polymer films, the observed decreasing frequency shifts of the thinner layers were also characteristic for the mass effect. The relatively heavy water molecules were imbibed in the absorption centers of the porous rr-P3HT polymer film with profile roughness Ra ~76 nm and RMS ~94 nm estimated on the base of AFM measurement (Figure 4).

Humidity sensors were also fabricated by using thin porous receptor layers, in which the presence of pores have crucial influence on changing the properties of the receptor layer [15,19]. The sensitivity of such systems is dependent on the amount of water captured, thus it depends on the dimensions, volume and distribution of pores in such a receptor layer. The presence of π-conjugated polymer chains of the rr-P3HT compound promotes sorption of humidity via interactions with the polar water molecules. This phenomenon also results in increasing of the mean diffusion path of the water molecules through the receptor layer and the magnitude of this effect is proportional to the thickness of the receptor active film. At higher relative humidity levels, water was likely condensing within the pores, hindering the desorption of humidity from the receptor layer and thus decreasing the surface of the receptor layer available to the analyte. The comparison of the characteristics of some SAW humidity sensing structures are presented in Table 3.

In the range of humidity concentrations as low as ~5% rh, the response time of the rr-P3HT-based sensors is higher or comparable to that of other structures [15,16,17,19], but at the expense of their lower sensitivity.

## 5. Conclusions

The rr-P3HT polymer films were utilized in a single (LiNbO_3_) and dual (quartz) SAW delay line configurations for humidity detection. The larger phase shifts in the LiNbO_3_-based sensors were observed for the 3rd harmonic SAW in comparison to the 1st ones, which was characteristic of the mass effect; decreasing sensor responses were observed with decreasing thickness of the sensing film. As a consequence, a thicker rr-P3HT film was applied (~600 nm) in the quartz-based SAW dual-delay line configuration which enabled the humidity sensing in a low-medium range with a linear calibration curve and a satisfactorily good sensitivity of ~−20.5 Hz/% rh.

The simple and cheap spray coating deposition technology for rr-P3HT polymer films, complemented with very fast (~5 s response time much faster than commercially available Michell SF-52 device) and low level humidity detection (~5% rh) in SAW dual-delay line system, highlights their potential in a low-medium range humidity sensing applications.

## Figures and Tables

**Figure 1 sensors-24-03651-f001:**
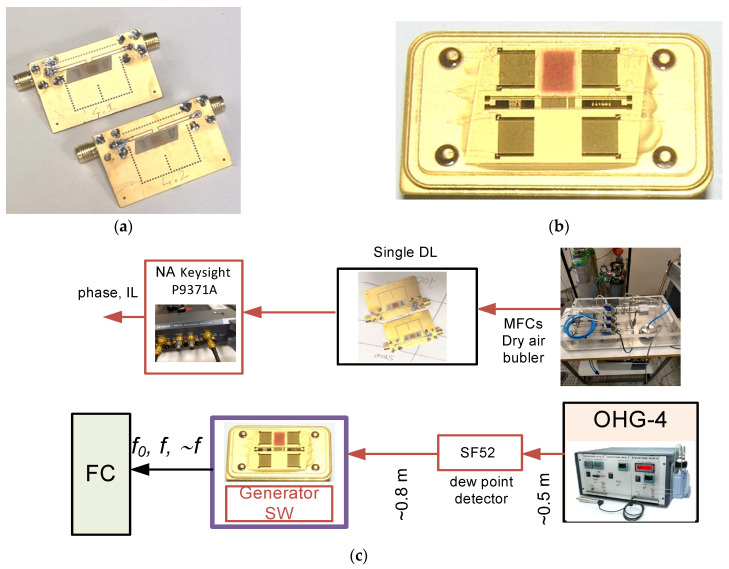
The images of the SAW devices (including the packaging and rr-P3HT films) onto (**a**) 128^0^ Y-X LiNbO_3_ and (**b**) ST-x quartz substrates; (**c**) schematic diagram of the setup utilized for testing the single delay line (**top** picture) and dual delay lines (**bottom** picture) (FC frequency counter, SW switchable generator, OHG-4 Owlstone humidity generator, SF52 Michell dew point detector, MFC mass flow controllers, NA network analyzer).

**Figure 2 sensors-24-03651-f002:**
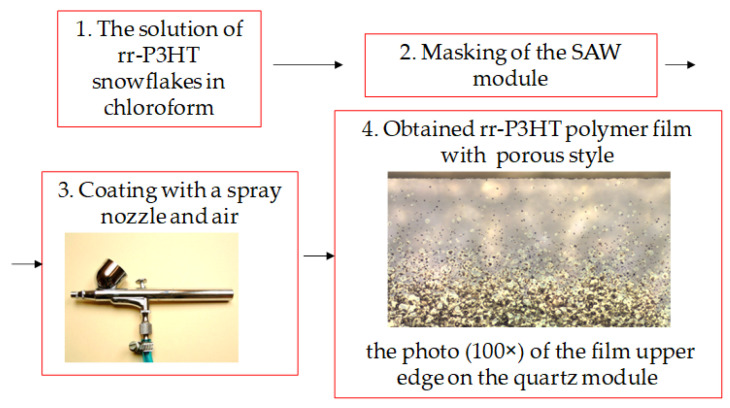
The individual four steps of the spray coating method for manufacturing of rr-P3HT polymer films.

**Figure 3 sensors-24-03651-f003:**
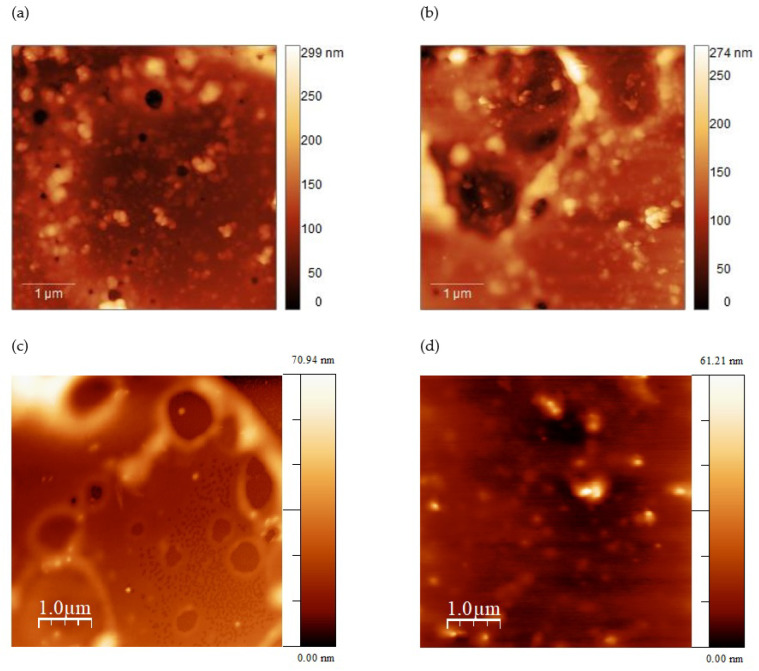
The AFM measurements in tapping mode of the rr-P3HT films with (**a**) ~240 nm, (**b**) ~130 nm, (**c**) ~80 nm and (**d**) ~40 nm, deposited onto the single delay line on LiNbO_3_ substrate.

**Figure 4 sensors-24-03651-f004:**
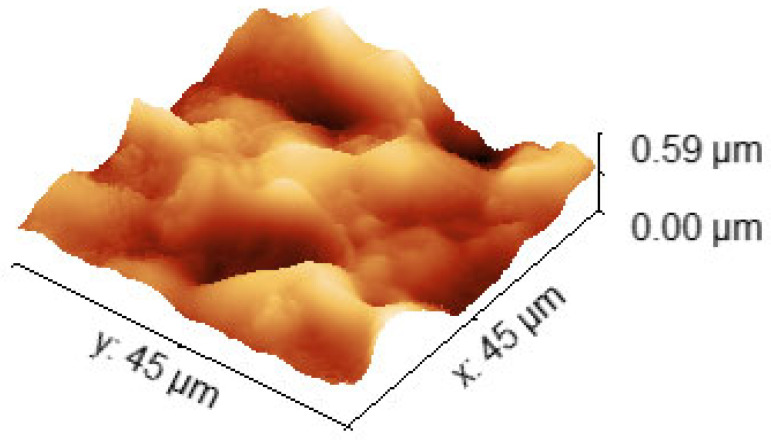
AFM measurement of the rr-P3HT thick film (~600 nm) deposited onto the quartz dual-delay line module; its profile roughness Ra ~76 nm and RMS ~94 nm.

**Figure 5 sensors-24-03651-f005:**
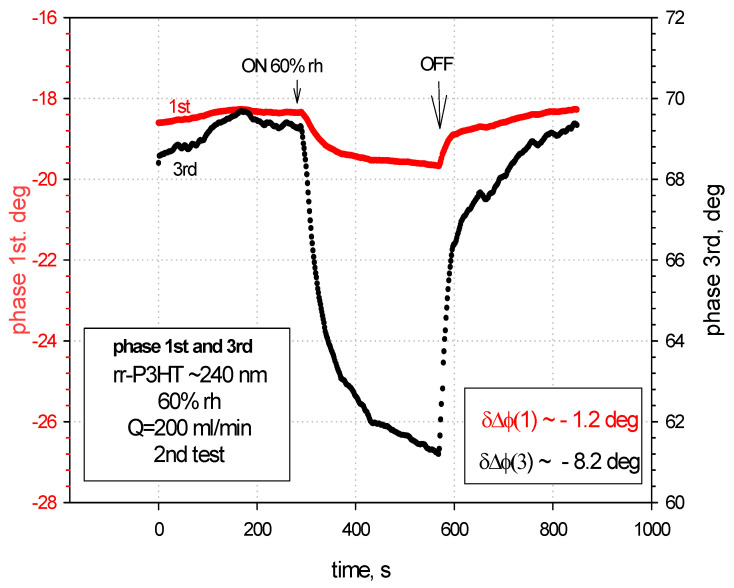
The phase shifts for the SAW 1st (red curve) and 3rd (black curve) harmonics with rr-P3HT ~240 nm thick at 60% rh.

**Figure 6 sensors-24-03651-f006:**
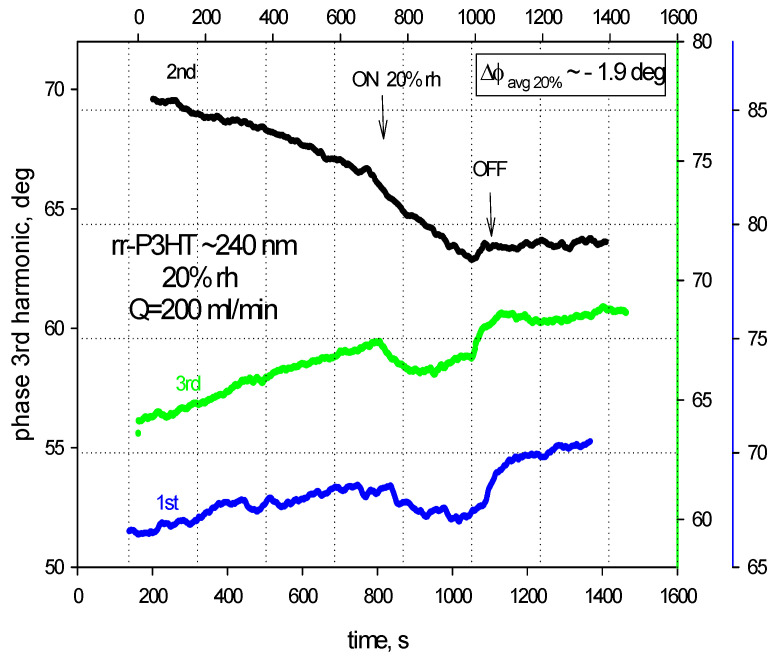
The phase shifts of the SAW 3rd harmonic (~243 MHz) for the rr-P3HT polymer film ~240 nm thick at 20% RH in a single line on LiNbO_3_ substrate.

**Figure 7 sensors-24-03651-f007:**
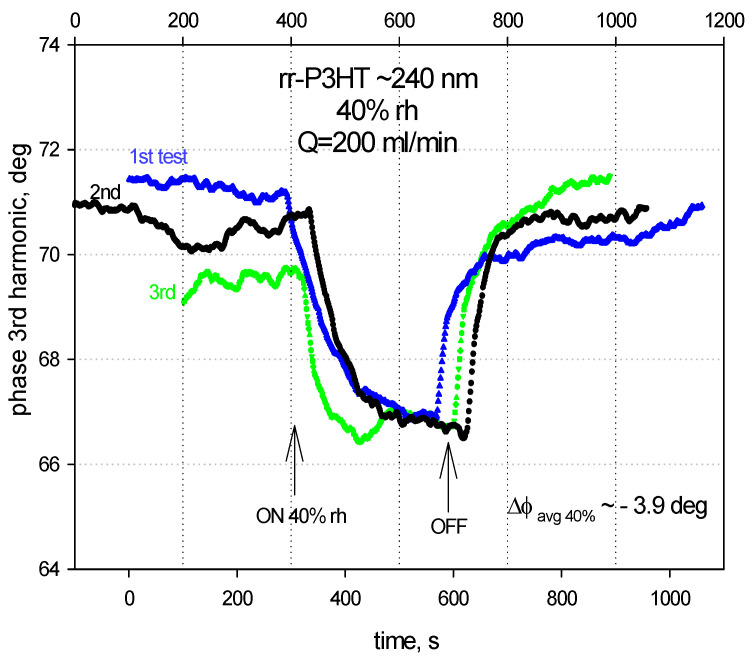
The phase shifts of the SAW 3rd harmonic (~243 MHz) for the rr-P3HT polymer film ~240 nm with 40% rh in a single line on LiNbO_3_ substrate.

**Figure 8 sensors-24-03651-f008:**
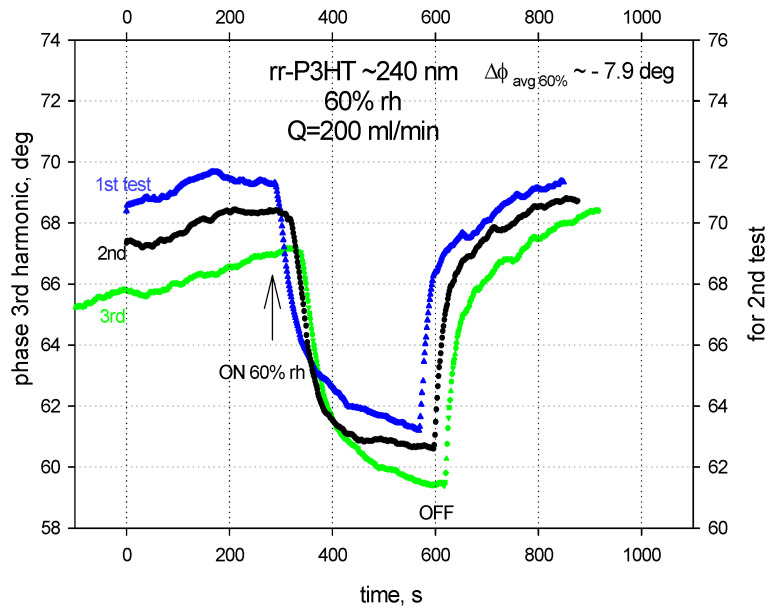
Phase shifts of the SAW 3rd harmonic (~243 MHz) for the rr-P3HT polymer film ~240 nm and 60% rh in a single line on LiNbO_3_ substrate.

**Figure 9 sensors-24-03651-f009:**
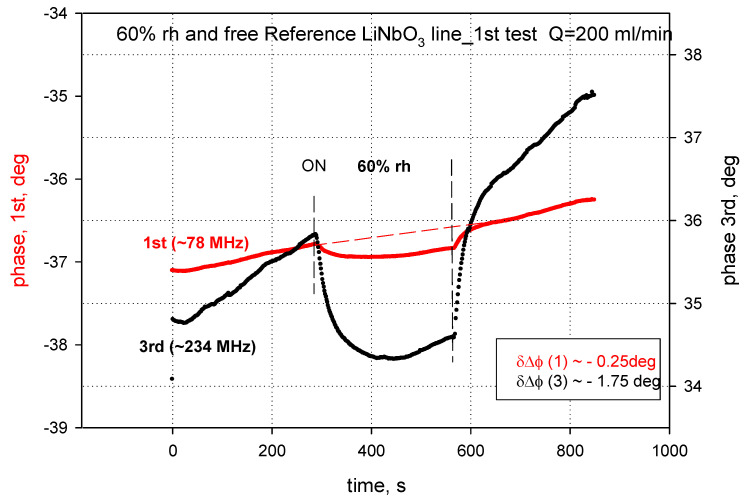
Interactions of the for the SAW 1st and 3rd harmonics travelling along the bare LiNbO_3_ surface in wet air (RH = 60%).

**Figure 10 sensors-24-03651-f010:**
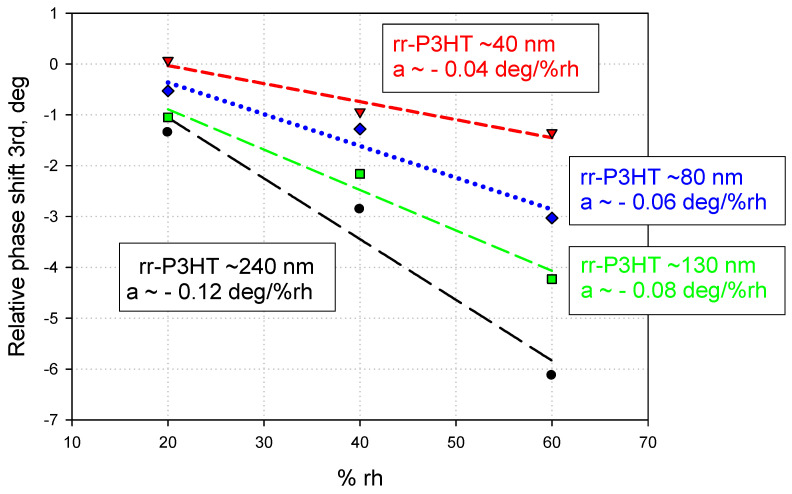
Absolute (relative to the reference free path) phase shifts for rr-P3HT films versus the RH in the 20–60% range for the SAW 3rd harmonic, for different sensing layer thicknesses; RPSe (Relative Phase Sensitivity) = Δϕ (rr-P3HT) − Δϕ(ref.).

**Figure 11 sensors-24-03651-f011:**
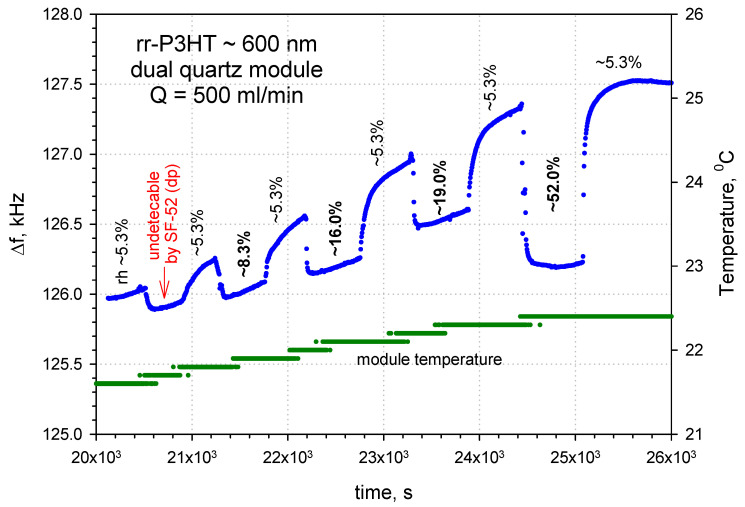
Frequency shifts vs time curve for the SAW dual delay line sensor on quartz with rr-P3HT polymer film ~600 nm thick at different humidity levels.

**Figure 12 sensors-24-03651-f012:**
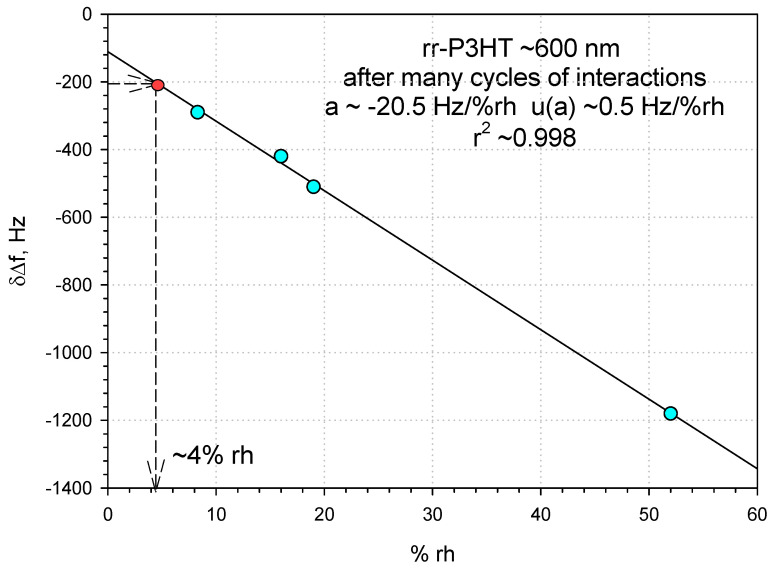
The frequency shifts δΔf = Δf (due various % rh) − Δf_0_ (due to the base line 5.3% rh) versus rh% for the rr-P3HT polymer film ~600 nm thick in a quartz dual delay line system.

**Table 1 sensors-24-03651-t001:** Phase shifts (deg) of the 3rd harmonic (averaged from 3 tests) on LiNbO_3_ covered by rr-P3HT films.

Thickness *, nm	20% rh	40% rh	60% rh
0, free reference line	−0.55	−1.04	−1.77
~240	−1.90	−3.90	−7.90
~130	−1.60	−3.20	−6.00
~80	−1.08	−2.32	−4.80
~40	−0.48	−1.98	−3.12

* estimated values from the AFM measurements.

**Table 2 sensors-24-03651-t002:** Relative Phase shifts (deg) of the 3rd harmonic on LiNBO_3_ covered by rr-P3HT films.

Thickness *, nm	20% rh	40% rh	60% rh
~240	−1.35	−2.86	−6.13
~130	−1.05	−2.16	−4.23
~80	−0.53	−1.28	−3.03
~40	+0.07	−0.94	−1.35

* estimated values from the AFM measurements.

**Table 3 sensors-24-03651-t003:** Summary of the characteristics of recent SAW humidity sensing structures.

Material	Sensitivity	Response/Recovery Time	Reference
SnO_2_/MoS_2_	0.78 kHz/% rh	~100 s/100 s	[17]
polymer PVA (polyvinyl alcohol)	3.7 kHz/% rh	~30–35 s/~40 s	[15]
Al_2_O_3_	8.7 kHz/% rh	50 s/50 s	[16]
MoS_2_/GO	114 ppm/% rh above 20% rh	6.6 s/3.5 s above 20% rh	[19]
Michell SF-52 (dew point)	-	~minutes at ~5% rh/~minutes	commercially available
rr-P3HT	~20.5 Hz/% rh	~5 s/~5 s at 5% rh	this work

## Data Availability

Data are contained within the article and Appendix A.

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
