# Peer review of "SAW Humidity Sensing with rr-P3HT Polymer Films"

_sensors, 2024, doi:10.3390/s24113651_

Round 1
Reviewer 1 Report
Comments and Suggestions for Authors
This paper reported a surface acoustic wave (SAW) humidity sensor based on P3HT (poly-3-hexylthiophene) polymer films via spray coating technology. The sensor exhibits a good performance. However, some specific problems should be addressed.
1、Please give a physical diagram for the sensor measurement.
2、Higher harmonics show better sensitivity, why not use the test results of higher harmonics as experimental data,such as 5th or 7th harmonics.
3、In Figure 8, there are two variables, please give the response diagram under the same temperature and different humidity, or the response diagram under the same temperature with different humidity.
4、In Figure 8, please explain in detail the reason for the frequency drift.
5、How about the humidity hysteresis of the sensor?
Author Response
1、Please give a physical diagram for the sensor measurement. - Yes, the suitable diagram has been included in the revised version as Figure 1c
2、Higher harmonics show better sensitivity, why not use the test results of higher harmonics as experimental data,such as 5th or 7th harmonics. - the IDTs used in the present paper work well at the fundamental and third harmonic Rayleigh waves, while the fifth and seventh harmonics are excited as well but are highly attenuated thus they are not suitable for further sensing tests - the suitable figure "disp 4.1 freq" in attachement.
3、In Figure 8, there are two variables, please give the response diagram under the same temperature and different humidity, or the response diagram under the same temperature with different humidity. -
The figure 8 there is the revised version as Figure 11, the symbol T means the dew point temperature on the base of the SF 52 detector - the rh% is suitable recalculated value - it is explained in the figure 11 caption.
4、In Figure 8, please explain in detail the reason for the frequency drift. - The frequency drift is a consequence of the slowly (long-time) increasing temperature inside the measuring chamber (for example 18.0 - 21.5 because of the electronic elements working, also the external temperature increase).
5、How about the humidity hysteresis of the sensor?
The hysteresis will be detailed tested in the future research, however, on the base of the current investigations it seems to be in a reasonable range even at low humidity range ~5% rh.

Reviewer 2 Report
Comments and Suggestions for Authors
1,In Introduction, “their often mutual drawback is the low accuracy and hysteresis effect at the low humidity detection levels”. The author should cite the more recent literature for explanation.
2,The author should take a separate picture of the SAW sensors in Figure 1, and label the size and structure.
3,In, page2 line 77-78. Please explain why choice the distance is 40mm?
4,Figure 3 Only two pictures, please check?
5,Figure 4 , Figure 8 is not very clear, please change the clear picture.
6,Some formatting errors, first line indented incorrectly. Line 71, 126,147,164,174,179,191
Comments on the Quality of English Language
English needs to be checked in detail, and some sentences need to be reorganized.
Author Response
1,In Introduction, “their often mutual drawback is the low accuracy and hysteresis effect at the low humidity detection levels”. The author should cite the more recent literature for explanation. Yes, the much more recent literature are cited in the revised version - the introduction section is extended.
2,The author should take a separate picture of the SAW sensors in Figure 1, and label the size and structure. Yes, the additional scheme of the SAW configurations is presented as Figure 1c
3,In, page2 line 77-78. Please explain why choice the distance is 40mm?
On the base of the experimental practice, the most repeatable films were obtained just for the distance of ~40 mm, it is connected with the parameters of the spray coating preasure and the nozzle diameter. Because a distance of less than ~ 40 mm causes (for this scale for a SAW substrate) too strong a blow of the atomized solution on the transducer, which can cause the agent to penetrate under the mask and settle on the IDT transducers, resulting in its permanent damage. A distance of more than 40 mm leads to unnecessary and lossy covering of a larger area of the mask with a solution of rrP3HT polymer. - it is added to the revised version
4,Figure 3 Only two pictures, please check? Please Sorry, Yes it is improved in revised version.
5,Figure 4 , Figure 8 is not very clear, please change the clear picture. Have been improved - in Figure 8 (in revised version Figure 11) T means the dew point temperature of the SF53 detector - explained in the text
6,Some formatting errors, first line indented incorrectly. Line 71, 126,147,164,174,179,191, OK, Yes it is all improved.
Reviewer 3 Report
Comments and Suggestions for Authors
It can be said that this is a report, not an article, and the experimental results can only basically support the conclusion. A major revision is recommended.
1. The Introduction part is too simple and lacks the description of other people's relevant work, as well as the description of innovation, and the word count is seriously insufficient.
2. Lack of characterization and morphology of material composition, such as SEM, TEM,XPS, etc.
3. The relevant reagents of the preparation materials used are not listed.
4. The testing method of the equipment is not specified, such as AFM sample preparation.
5. Why not test for high humidity, such as 80%RH?
6. Research efforts lack comparison with peers.
Author Response
- The Introduction part is too simple and lacks the description of other people's relevant work, as well as the description of innovation, and the word count is seriously insufficient.
YES, it has been improved by including more sentences and suitable literature.
- Lack of characterization and morphology of material composition, such as SEM, TEM,XPS, etc.
The morphology of the polymer film is tested by means of AFM technic which is sufficient for assessing of the polymer films sensor properties
- The relevant reagents of the preparation materials used are not listed. All utilized materials were prepared with very clean properties, especially the new spray coating method is utilizing very clean dry air from the bottle.
- The testing method of the equipment is not specified, such as AFM sample preparation. The more detailed description is included in the revised version. AFM were performed in a standard tapered mode (non-contact)
- Why not test for high humidity, such as 80%RH? It was tested, in such a ranges there exist many very good detectors, however, the proposed solution also work quite good and will be showed in another paper.
- Research efforts lack comparison with peers. The proposed SAW configuration with thin rrP3HT polymer films is in a position to detect humidity below 5% rh much faster than even the professional Michell SF52 device with dew point detector.
Reviewer 4 Report
Comments and Suggestions for Authors
1) The article has insufficient references. The introduction section is too generic and does not reference any articles to highlight issues that this paper is trying to address. I suggest re-writing the introduction to elucidate the problem more clearly.
2) It is not clear how the sensors were fabricated or assembled. A schematic showing the process written in section 2 would be useful.
3) The image of the spray coating nozzle adds no value to this paper. Also, it is not clear what the other image in Figure 2 is. Please add a scale and a zoomed out image to show what the reader is looking at.
4) Figure 3c and 3d are missing in the manuscript. Figure 3a and 3b are missing a z-axis scale. It is not clear what the AFM images are showing. Is there a step height that is being measured?
5) For section 3.2, please describe the experimental setup and how these tests are carried out. Is the response of the device to a pulse of water vapor? How is the relative humidity measured at 60%?
6) Is the drift or shift in baseline expected in 1st/2nd/3rd test in Figure 5?
7) This paper needs a lot of correction for formatting and English language.
8) How does this sensor compare to the state of the art? It is not clear if the sensor is able to demonstrate fixing the problems that this paper set out to address.
Comments on the Quality of English Language
Please have this paper edited. It needs extensive formatting and English correction.
Author Response
1) The article has insufficient references. The introduction section is too generic and does not reference any articles to highlight issues that this paper is trying to address. I suggest re-writing the introduction to elucidate the problem more clearly.
Yes the references are much improved and the Introduction section is extended.
2) It is not clear how the sensors were fabricated or assembled. A schematic showing the process written in section 2 would be useful.
Yes the schematic diagram is included as Figure 1c.
3) The image of the spray coating nozzle adds no value to this paper. Also, it is not clear what the other image in Figure 2 is. Please add a scale and a zoomed out image to show what the reader is looking at.
Yes Figure 2 is improved with the more understandable captions.
4) Figure 3c and 3d are missing in the manuscript. Figure 3a and 3b are missing a z-axis scale. It is not clear what the AFM images are showing. Is there a step height that is being measured?
Please sorry, in pdf file the 3c and 3d were cutted, Yes it has been improved, the figure has now the z-axis
5) For section 3.2, please describe the experimental setup and how these tests are carried out. Is the response of the device to a pulse of water vapor? How is the relative humidity measured at 60%?
Yes it has been described more precisely, Yes the response are for the humidity pulses, measured by means of SF52 detector
6) Is the drift or shift in baseline expected in 1st/2nd/3rd test in Figure 5?
Yes the slight drift is expected what is visible in Figure 5, the axis for 2nd test is on the right side.
7) This paper needs a lot of correction for formatting and English language.
Yes the correction and formatting has been applied.
8) How does this sensor compare to the state of the art? It is not clear if the sensor is able to demonstrate fixing the problems that this paper set out to address.
The proposed solution can be compared with devices with fast response especially at low humidity ranges <5% because the commercially available detector Michell SF52 is undetectable such a low water concentration in a reason time (even in several minutes).
Round 2
Reviewer 1 Report
Comments and Suggestions for Authors
In lines 49-50 of page two, the author said that the mass effect depends on SAW frequency, thus utilization of higher harmonics in phase measurements is highly indicated. please detailly illustrate why not use frequency measurement to instead of phase?
The clarity of Fig. 11 sohuld be modified, including some characters. what dose mean of δ△f. Please check the entire paper clearly.
Author Response
In lines 49-50 of page two, the author said that the mass effect depends on SAW frequency, thus utilization of higher harmonics in phase measurements is highly indicated. please detailly illustrate why not use frequency measurement to instead of phase?
In the paper the both (phase and frequency) methods were utilized. The phase measurement were used for initial testing of the humidity properties of the rr-P3HT polymer films, than the final results were performed by means of frequency method (in dual delay line configurations) with the frequency ( ~205 MHz) comparable to the 3rd harmonic (~234 MHz) of the phase measurement.
The clarity of Fig. 11 should be modified, including some characters. what dose mean of δ△f. Please check the entire paper clearly.
Yes, Fig.11 is modified for the better understanding. The symbol δ△f means the difference between the value of the signal △f under humidity dosing versus his initial value △f0 (on the level of base line humidity), so the δ△f = △f - △f0 .
Yes, the entire paper has been checked.
Reviewer 2 Report
Comments and Suggestions for Authors
It seems that the quality of Figure 4 and Figure 8 ( in revised version Figure 11) has not improved and are still not clear, please provide clearer figures, not the text.
Comments on the Quality of English Language
Minor editing of English language required
Author Response
It seems that the quality of Figure 4 and Figure 8 ( in revised version Figure 11) has not improved and are still not clear, please provide clearer figures, not the text
Yes, the quality has been improved, Figure 4 is enlarged and the Figure 11 has the better quality and is more clear for understanding.
Reviewer 3 Report
Comments and Suggestions for Authors
- I still think the morphology of the polymer film is tested by means of AFM technic which is not sufficient for assessing of the polymer films sensor properties.
- Research efforts still lack comparison with peers. Please list them included other research works in a table.
- How did the authors prepared the sensing materials?
Author Response
I still think the morphology of the polymer film is tested by means of AFM technic which is not sufficient for assessing of the polymer films sensor properties.
At the current stage of the study, the authors have planned only the AFM measurements for the samples prepared by means of the spray coating technology and utilized in SAW measurements. The AFM results made it possible to estimate the average thickness of the obtained layers.
The main goal of the paper, i.e. the sensor properties of the prepared polymer films (vs. the air humidity various levels) are assessed in the SAW phase and frequency precise investigations, presenting considerable phase and frequency shifts with the very fast responses (~5 s especially in the frequency configurations) even at low humidity levels ( near 5% rh).
The AFM description has been improved (below) and placed in body text in 3.1:
Figures 3 and 4 show AFM images data of the surface topography of the rr-P3HT layers deposited on LiNbO3 and quartz substrates. Fig. 3 (panels a to d) shows the 2D AFM maps collected at a scan-size of 5 μm referred to samples having thicknesses of ~240, ~130, ~80 and 40 nm, respectively, which were used in films in a single delay line configuration. The corresponding Rs surface roughness values are 30 nm, 28 nm, 10 nm, and 5 nm. The AFM topography of a much thicker film (~600 nm) used in a dual-delay line quartz module for the more precise frequency investigations, is reported in Fig.4 showing the 3D rendering of the morphology. The Rs value for this sample is ~94 nm. The AFM data revealed that the surface roughness of the layers increases with increasing the layer thickness. The morphology data of Fig. 3 and Fig. 4 show that the roughness is mainly due to the characteristic features of the thin spray-coated films. In fact, circular holes or pits are present, which are likely originated by the evaporation of the solvent present in the film as it is being deposited. The lateral size of such structures is in the range between few hundreds of nanometers and about 1 mm, while the depth is of the order of 10 nm. These structures are clearly less present and pronounced for the thinnest film (~40 nm).
Research efforts still lack comparison with peers. Please list them included other research works in a table.
Yes, the additional Table 3 is included to the manuscript body presenting “Comparison of the characteristic of the recent SAW humidity sensors” during the last 5 years.
Table 3. Summary of the characteristics of recent SAW humidity sensing structures
|
Material |
Sensitivity |
response/recovery time |
reference |
|
SnO2/MoS2 |
0.78 kHz/ %rh |
~100 s/ 100 s |
[17] |
|
polymer PVA (polyvinyl alcohol) |
3.7 kHz/ %rh |
~30-35 s/ ~40 s |
[15] |
|
Al2O3 |
8.7 kHz/ %rh |
50 s / 50 s |
[16] |
|
MoS2/GO |
114 ppm/%rh above 20%rh |
6.6 s / 3.5 s above 20%rh |
[19] |
|
Michell SF52 (dew point) |
- |
~ minutes at ~5%rh/ ~minutes |
commercially available |
|
rr-P3HT |
~20.5 Hz/ %rh |
~5 s/ ~5 s at 5% rh |
this work |
In the range of low humidity concentrations ~5% rh, the rate of action of polymeric structures of the rr-P3HT type is higher or comparable than other structures [15,16,17,19], however, at the expense of their lower sensitivity.
How did the authors prepared the sensing materials?
The details of the sensing materials preparation are added in the Supplement Materials
“Materials. 2,5-dibromo-3-hexylthiophene (TCI, ≥97%), t-butylmagnesium chloride (Sigma Aldrich, 2M ether solution), dichloro-[1,3-bis(diphenylphosphino)propane]nickel(II) (Ni(dppp)Cl2) (Sigma Aldrich, ≥97%) were used as received, without further purification. Anhydrous tetrahydrofurane (ACROS Organics, 99.9%) was distilled over metallic sodium prior to use. All reactions were conducted under dry nitrogen or argon flow, in oven-dried glassware.
Regioregular PHT was synthesized via the McCullough GRIM method (doi: 10.1016/j.polymer.2005.05.035), modified as per the work of De Girolamo (doi: 10.1021/jp0741758): a dry 100 cm3 three-necked flask, equipped with septum, a condenser, a gas capillary and a magnetic dipole, was purged with nitrogen and charged, via syringe, with 2,5-dibromo-3-hexylthiophene (6.13 mmol), anhydrous tetrahydrofurane (21.5 cm3) and t-butylmagnesium chloride (7.46 mmol). The reaction mixture was refluxed for 2 h, followed by addition of the Ni(dppp)Cl2 catalyst (0.0318 mmol) and heating for the next 1 h. The crude polymer was precipitated by quenching the reaction mixture in methanol.
Purification of crude products. Obtained polymer were purified with sequential Soxhlet extraction with methanol, hexane and chloroform. Vacuum-dried chloroform fractions were characterized by means of 1H-NMR and SEC analyses and used as such in humidity sensing investigations.
Molecular characterisation of obtained products. 1H-NMR analysis was performed from solutions in CDCl3 on a Varian Unity Inova ( Palo Alto, CA, USA) spectrometer with a resonance frequency of 300 MHz using TMS as internal standard. The number average molecular weights and dispersities were determined using a size-exclusion (SEC) chromatograph, equipped with an 1100 Agilent 1260 Infinity isocratic pump, an autosampler, a degasser, a thermostatic box for columns and a differential refractometric MDS RI Detector (Santa Clara, CA, USA). The molecular weight obtained by SEC was based on calibration with linear polystyrene standards (580- 300,000 g/mol). Pre-column guard 5 μm (50 x 7.5 mm) and PLGel 5 μm MIXED-C (300 x 7.5 mm) column were used for separation. The measurements were carried out in THF (HPLC grade) as the solvent, at 30°C with a flow rate of 0.8 cm3/min.
RR-PHT: 1H-NMR (CDCl3, 300 MHz) δH, ppm: 6.98 (s, 1H), 2.81 (m, 2H), 1.76-1.66 (m, 2H), 1.47-1.34 (m, 6H), 0.91 (t, J=6,9 Hz, 3H).
RR-PHT have been prepared, with molecular weights of 10,000 g/mol, and dispersities of 1.3 respectively (as determined by size exclusion chromatography, SEC).